# Comparative Analysis of Mouse Decidualization Models at the Molecular Level

**DOI:** 10.3390/genes11080935

**Published:** 2020-08-13

**Authors:** Chong Wang, Miao Zhao, Wen-Qian Zhang, Ming-Yu Huang, Can Zhu, Jia-Peng He, Ji-Long Liu

**Affiliations:** 1Guangdong Provincial Key Laboratory of Agro-Animal Genomics and Molecular Breeding, National Engineering Research Center for Breeding Swine Industry, College of Animal Science, South China Agricultural University, Guangzhou 510642, Guangdong, China; bettyscau@126.com (C.W.); zm1586186423@126.com (M.Z.); 2Guangdong Laboratory for Lingnan Modern Agriculture, College of Veterinary Medicine, South China Agricultural University, Guangzhou 510642, Guangdong, China; zhangwqian@126.com (W.-Q.Z.); hmingyu402@163.com (M.-Y.H.); yuliaohui@126.com (C.Z.); stevencapeng@sina.com (J.-P.H.)

**Keywords:** decidualization, RNA-seq, uterus, mouse

## Abstract

The mouse is widely used to study decidualization and there are three well-established mouse models of decidualization, namely natural pregnancy decidualization (NPD), artificial decidualization (AD), and in vitro decidualization (IVD). However, the extent of similarity and difference between these models at the molecular level remains largely unknown. Here, we performed a comparative analysis using the RNA-seq approach. In the NPD model, which is thought to be the golden standard of mouse decidualization, we found a total of 5277 differentially expressed genes, with 3158 genes being up-regulated and 2119 genes being down-regulated. A total of 4294 differentially expressed genes were identified in the AD model: 1127 up-regulated genes and 3167 down-regulated genes. In comparison to NPD, 1977 genes were consistently expressed, whereas only 217 genes were inconsistently expressed, indicating that AD is a reliable model for mouse decidualization. In the IVD model, RNA-seq analysis revealed that 513 genes were up-regulated and 988 genes were down-regulated. Compared to NPD, 310 genes were consistently expressed, whereas 456 genes were inconsistently expressed. Moreover, although the decidualization marker Prl8a2 (prolactin family 8 subfamily a member 2) was up-regulated, the widely-used marker Alpl (alkaline phosphatase liver/bone/kidney) was down-regulated in the IVD model. Therefore, we suggest that the IVD model should be optimized to mimic NPD at the transcriptomic level. Our study contributes to an increase in the knowledge about mouse models of decidualization.

## 1. Introduction

Endometrial stromal cell decidualization is a prerequisite for embryo implantation and pregnancy in human reproduction [1]. Decidualization is a spontaneous process, which initiates in the secretory phase of the menstrual cycle controlled by ovarian steroid hormones [2]. During this process, endometrial stromal cells change into large epithelioid cells and secrete two protein markers, decidual prolactin (dPRL) and insulin-like growth factor-binding protein 1 (IGFBP1) [3]. Decidualization plays important roles in embryonic implantation, placentation, and pregnancy maintenance [4]. Insufficient decidualization may lead to repeated pregnancy loss (RPL), intrauterine growth restriction (IUGR), and severe pre-eclampsia (sPE) [5].

Due to ethical restrictions, direct probing of decidualization in humans is difficult and the mouse is widely used to study human decidualization. Decidualization in mice is embryo-dependent, which is slightly different from decidualization in humans [2]. Prl8a2 (prolactin family 8 subfamily a member 2) and Alpl (alkaline phosphatase liver/bone/kidney) are marker genes for decidualization in mice. Shortly after embryo implantation on day 5 of pregnancy, anti-mesometrial stromal cells in the vicinity of the implanted embryo undergo decidualization, which is called the primary decidual zone (PDZ). The decidualization reaction rapidly extends to the entire anti-mesometrial region, when the secondary decidual zone (SDZ) is formed [6]. On day 8 of pregnancy, the uterus, from the anti-mesometrial region to the mesometrial region, is fully decidualized. Interestingly, uterus primed by ovarian steroid hormones can be artificially decidualized by mechanical means (e.g., sesame oil). The artificially decidualized uterus is morphologically similar to the uterus during natural pregnancy, making it a useful decidualization model free of embryonic factors [7]. Additionally, it has been shown that decidualization can be triggered in isolated endometrial stromal cells by estrogen plus progesterone [8]. However, the extent of similarities/differences between these mouse models of decidualization at the molecular level remains largely unknown.

In this study, by using RNA-seq, we investigated the global gene expression profiles of 3 mouse models of decidualization: natural pregnancy decidualization on day 8 of pregnancy (NPD), artificial decidualization of mouse uterus stimulated by sesame oil (AD), and in vitro decidualization of cultured stromal cells by incubation with estrogen plus progesterone (IVD). RNA-seq is more accurate than the microarray for quantifying gene expression profile, because it detects novel transcripts [9], discriminates very similar sequences [10], and extends the quantification limit [11]. Our study may provide a valuable resource for further studies on the molecular mechanism of decidualization.

## 2. Materials and Methods

### 2.1. Mouse Models of Decidualization

The natural pregnancy decidualization (NPD) model was established by mating fertile females with fertile males of the CD-1 strain. The day of the vaginal plug was noted as day 1. On day 8 of pregnancy, implantation sites (decidualized) and non-implantation sites (served as a control) of the uterus were collected separately. Embryonic tissues at the implantation site were removed under a stereomicroscope. In addition to NPD, the artificial decidualization (AD) model was established by co-caging female mice with vasectomized males to become pseudo-pregnancy. On day 4 of pseudo-pregnancy, 20 μL sesame oil was injected into one of the uterine horns and the unstimulated uterine horn served as the control. Uterine samples were obtained on day 8 of pseudo-pregnancy. For in vitro decidualization (IVD) model, mouse uterus was collected on day 4 of pseudo-pregnancy. Endometrial stromal cells were isolated from mouse uterus as described previously [12]. To induce in vitro decidualization, cells were treated with 10 nM estradiol-17β and 1 μM progesterone. Cells were harvested on day 4 of IVD. All animal procedures were approved by the Institutional Animal Care and Use Committee of South China Agricultural University.

### 2.2. RNA-seq

We extracted total RNAs from uterine samples with the TRIzol reagent (Invitrogen, Carlsbad, CA, USA). The ND-1000 Nanodrop, as well as the Agilent 2200 TapeStation was employed to assess the purity and integrity of total RNAs. The quality control parameters used in this study were: A260/A280 ratio ≥1.8, A260/A230 ratio ≥2.0, and RNA integrity number ≥8.0. The TruSeq RNA sample preparation kit (Illumina, San Diego, CA, USA) was used to generate cDNA libraries. High-throughput sequencing was run on an Illumina HiSeq 2500 system. Raw RNA-seq data were processed with an in-house computational pipeline as described previously [13]. Briefly, clean reads were mapped to the mouse genome (UCSC mm9) with the TopHat software v2.0.4 [14] and then assembled at the gene level with the Cufflinks software v2.2.1 [15]. Genes with fold change ≥2 and false discovery rate (FDR) ≤0.01 were chosen as differentially expressed genes. The RNA-seq raw data were deposited in Gene Expression Omnibus (GEO) with the accession number GSE122376.

### 2.3. Validation by Quantitative RT-PCR

We used the TRIzol reagent (Invitrogen, Carlsbad, CA, USA) to extract total RNAs. The cDNAs were synthesized with the PrimeScript reverse transcriptase reagent kit (TaKaRa, Dalian, China). Quantitative RT-PCR was performed on Applied Biosystems 7500 (Life Technologies, Carlsbad, CA, USA) with the THUNDERBIRD SYBR qPCR Mix (Toyobo, Osaka, Japan). The glyceraldehyde-3-phosphate dehydrogenase (Gapdh) gene was used as the reference gene for data normalization. A complete list of primer sequences is provided in Appendix A.

### 2.4. Gene Ontology (GO) Analysis

GO analysis was performed using a PERL script configured with the MGI GOslim database [16]. Only GOslim terms under biological processes were considered. Multiple test correction was not used because there were merely 14 biological process terms in total. The significance cutoff of 0.05 was used.

### 2.5. Pathway Analysis

The DAVID online tool was employed for pathway analysis as described previously [17]. FDR (false discovery rate) ≤ 0.05 was used as a significance cutoff.

### 2.6. Gene Network Analysis

The gene network was reconstructed by using the STRING online tool v10.0 [18]. The threshold score for gene-gene interaction was 0.4 by default. The Cytoscape tool [19] was used to view the network. Network Analyzer [20] was used to compute the degree distribution. The mean + 2 × SD (standard deviation) was chosen as the cutoff value to select hub genes.

## 3. Results

### 3.1. Identification of Changed Genes Associated with Decidualization During Natural Pregnancy

To investigate changed genes associated with decidualization during natural pregnancy in mice, the decidualized implantation site, as well as the non-decidualized non-implantation site, was collected on day 8 of pregnancy (Figure 1A). Three replicates were prepared for each group. Through RNA-seq analysis, we identified a total of 5277 differentially expressed genes (fold change ≥ 2 and FDR ≤ 0.01), of which 3158 genes were up-regulated and 2119 genes were down-regulated (Figure 1B and Appendix A).

To validate our RNA-seq data, a panel of 14 genes with various fold changes was randomly selected. Quantitative RT-PCR (qRT-PCR) was run on an independent set of replicates (Figure 1C). Results from qRT-PCR and RNA-Seq were concordant (*r* = 0.962, *p* = 2.25 × 10^−12^), indicative of the high quality of the RNA-seq dataset.

### 3.2. Characterizing Differentially Expressed Genes by Gene Ontology and Pathway Analysis

According to gene ontology (GO), differentially expressed genes can be categorized into 14 biological processes: signal transduction (8.6%), cell adhesion (2.2%), cell-cell signaling (1%), DNA metabolism (1.8%), RNA metabolism (8.9%), protein metabolism (11%), other metabolic processes (10.4%), transport (8.3%), cell organization & biogenesis (7.8%), cell cycle & proliferation (5.5%), death (3.2%), stress response (4%), developmental processes (9.3%), and other biological processes (18%) (Figure 2A). Based on hypergeometric test, 11 out of these terms were significantly enriched for differentially expressed genes, including cell adhesion (*p* = 0.0131), DNA metabolism (*p* = 9.78 × 10^−6^), RNA metabolism (*p* = 0.0113), protein metabolism (*p* = 1.76 × 10^−10^), other metabolic processes (*p* = 1.92 × 10^−10^), transport (*p* = 7.42 × 10^−5^), cell organization & biogenesis (*p* = 1.25 × 10^−10^), cell cycle & proliferation (*p* = 1.01 × 10^−10^), death (*p* = 5.41 × 10^−5^), stress response (*p* = 0.00197), and developmental processes (*p* = 8.93 × 10^−7^). These data indicated that mouse decidualization might invokes a variety of genes participating in a wide range of biological processes.

Additionally, KEGG pathway analysis was carried out with the DAVID software. Enriched pathways were: focal adhesion (FDR = 1.51 × 10^−5^), oxidative phosphorylation (FDR = 5.12 × 10^−5^), glutathione metabolism (FDR = 1.86 × 10^−4^), regulation of actin cytoskeleton (FDR = 2.73 × 10^−4^), Rap1 signaling pathway (FDR = 2.73 × 10^−4^), PI3K-Akt signaling pathway (FDR = 0.00183), cell cycle (FDR = 0.00262), ECM-receptor interaction (FDR = 0.00293), FoxO signaling pathway (FDR = 0.00601), p53 signaling pathway (FDR = 0.0149), neurotrophin signaling pathway (FDR = 0.024), thyroid hormone signaling pathway (FDR = 0.027), purine metabolism (FDR = 0.0308), MAPK signaling pathway (FDR = 0.044), and endocytosis (FDR = 0.049) (Figure 2B).

### 3.3. Searching for Hub Genes Through Network Analysis

The network for differentially expressed genes was created by using the STRING online tool. The reconstructed network had 2114 genes with 9987 edges (Figure 3A). Further analysis showed that this network was a small-scale network with some high connected nodes known as hub genes (Figure 3B). Within this network, we found a total of 104 hub genes (Figure 3C). Because of their key positions in the network, these hub genes are supposed to be more important than the others.

### 3.4. Global Comparison with Artificial Decidualization (AD) Model

The AD model was established by stimulating a hormonally primed uterus with sesame oil (Figure 4A). Artificially decidualized uterine samples including 3 biological replicates were obtained and subjected to RNA-seq. A total of 4294 genes were differentially expressed upon AD, of which 3167 genes were down-regulated and 1127 genes were up-regulated (Figure 4B and Appendix A). Compared to NPD, 1114 down-regulated genes and 863 up-regulated genes were shared (Figure 4C and Appendix A).

We next focused on genes that were inconsistently expressed between AD and NPD. There were 211 genes down-regulated in AD but up-regulated in NPD, whereas there were only 6 genes up-regulated in AD but down-regulated in NPD (Figure 5A and Appendix A). To further characterize these inconsistently expressed genes, GO analysis was performed. As a result, we identified 4 enriched GO terms: cell adhesion (*p* = 0.00220), RNA metabolism (*p* = 0.000248), cell cycle & proliferation (*p* = 0.000639), and developmental processes (*p* = 0.00105) (Figure 5B). Furthermore, two representative genes, Ptgs2 and Esr1, which were up-regulated in NPD but down-regulated in AD, were validated by using qRT-PCR (Figure 5C).

### 3.5. Global Comparison with In Vitro Decidualization (IVD) Model

The IVD model was established using isolated mouse endometrial stromal cells (Figure 6A). RNA-seq analysis revealed that 1501 genes were differentially expressed upon IVD, of which 988 genes were down-regulated and 513 genes were up-regulated (Figure 6B and Appendix A). Compared to NPD, 117 down-regulated genes and 193 up-regulated genes were shared (Figure 6C and Appendix A).

There were 456 inconsistently expressed genes between IVD and NPD. Among them, 370 genes were down-regulated in IVD but up-regulated in NPD, whereas 86 genes were up-regulated in IVD but down-regulated in NPD (Figure 7A and Appendix A). Based on GO, 5 terms were significantly enriched among these inconsistently expressed genes, namely cell adhesion (*p* = 8.74 × 10^−5^), DNA metabolism (*p* = 0.00168), cell organization & biogenesis (*p* = 3.74 × 10^−9^), cell cycle & proliferation (*p* = 1.07 × 10^−11^), and developmental processes (*p* = 1.57 × 10^−9^) (Figure 7B). Finally, we validated the expression pattern of two representative genes, Alpl and Bmp2, which were up-regulated in NPD but down-regulated during IVD (Figure 7C).

## 4. Discussion

Mice are widely used as the animal model for studying decidualization in humans. In the present study, by using RNA-seq, we investigated global gene expression profiles of 3 well-established mouse decidualization models, namely natural pregnancy decidualization of mouse uterus on day 8 of pregnancy (NPD), artificial decidualization of mouse uterus stimulated by sesame oil (AD), and in vitro decidualization of cultured mouse endometrial stromal cells by incubation with estrogen plus progesterone (IVD). Our study might provide a valuable resource for understanding the molecular mechanisms of decidualization.

NPD is thought to be the golden standard of mouse decidualization. In this model, we identified a total of 5277 differentially expressed genes, of which 3158 genes were up-regulated and 2119 genes were down-regulated. Quantitative RT-PCR (qRT-PCR) analysis of randomly selected genes indicated that the RNA-seq data were of high quality. Gene ontology (GO) and pathway analysis was conducted to explore the function of differentially expressed genes. As a result, 11 GO terms and 15 pathways were significantly enriched. This finding indicated that mouse decidualization might invoke many genes with a variety of functions. To narrow down the list of genes associated with mouse decidualization, gene prioritization was performed by selecting hub genes in the gene network. With a defined cut-off value, we found 104 hub genes (92 up-regulated genes and 12 down-regulated genes). The hub genes are supposed to be more important than the others in the network and thus deserve further investigation.

The AD model is a useful model of in vivo decidualization free of embryonic factors [7]. We identified a total of 4294 differentially expressed genes in the AD model, of which 1127 genes are up-regulated and 3167 genes are down-regulated. In comparison to NPD, 863 up-regulated genes and 1114 down-regulated genes were shared. We observed a small portion of inconsistently expressed genes: 211 genes were down-regulated in AD but up-regulated in NPD, and 6 genes were up-regulated in AD but down-regulated in NPD. These inconsistently expressed genes were likely regulated by paracrine signals from the embryo [21,22]. We focused on two inconsistently expressed genes, Ptgs2 (prostaglandin-endoperoxide synthase 2) and Esr1 (estrogen receptor 1), which were up-regulated in NPD but down-regulated in AD. In mouse uterus, Ptgs2 was found to be expressed in the luminal epithelium and the underlying stroma at the embryo implantation site [23]. Female mice lacking Ptgs2 exhibited impaired implantation and decidualization [24,25]. Esr1 is activated by the sex hormone estrogen. At first, studies using Esr1-null uterus demonstrated that implantation but not decidualization was Esr1 dependent [26]. It was reported later that these Esr1-null mice still expressed a truncated Esr1 protein with partial transcription regulatory function [27,28]. Finally, it was shown that the complete loss of Esr1 in epithelial and stromal cells of the uterus failed experimentally induced decidualization [29]. The down-regulation of Ptgs2 and Esr1 may attenuate decidual reaction in AD to some extent compared to NPD. Nevertheless, two classical markers of decidualization, Prl8a2 (prolactin family 8 subfamily a member 2, also known as Dtprp: decidual/trophoblast prolactin-related protein) and Alpl (alkaline phosphatase liver/bone/kidney), were faithfully up-regulated in AD. Additionally, the number of consistently expressed genes is larger than that of inconsistently expressed genes between AD and NPD. Therefore, we conclude that AD is a reliable model for mouse decidualization.

The main advantage of the IVD is the ability to efficiently manipulate gene expression using overexpression vectors or siRNAs. Although IVD of human endometrial stromal cells is well-established, publications on IVD in mice are scarce. E_2_+P_4_ treatment induces IVD in 7–10 days in humans. However, in this study, we treated mouse endometrial stromal cells with E_2_ + P_4_ for only 4 days, because (a) the 4-day estrus cycle in mice is much shorter than the 28-day menstrual cycle in humans; (b) we isolated mouse endometrial stromal cells on day 4 of pseudo-pregnancy, decidualization occurs 4 days later in vivo, and most importantly (c) primary mouse endometrial stromal cells can only divide 2–3 times in vitro and a longer time of IVD causes abnormal morphological changes likely due to replicative senescence. RNA-seq analysis revealed that 1501 genes were differentially expressed upon IVD, of which 988 genes were down-regulated and 513 genes were up-regulated. Compared to NPD, 117 down-regulated genes and 193 up-regulated genes were shared. Unexpectedly, there were 456 inconsistently expressed genes between IVD and natural pregnancy: 370 genes were down-regulated in IVD but up-regulated in NPD, whereas 86 genes were up-regulated in IVD but down-regulated in NPD. Strikingly, although the decidualization marker Prl8a2 was up-regulated, the widely-used decidualization marker Alpl was down-regulated in IVD. In addition to mRNA, we also confirmed that the protein or even enzyme activity of Alpl was down-regulated in IVD (data not shown). Bmp2 encodes a secreted ligand of the transforming growth factor β superfamily of proteins. It has been reported that Bmp2-null uterine stroma failed to undergo decidualization [30,31]. Bmp2 was up-regulated in NPD but down-regulated during IVD. Other inconsistently expressed genes with a known function in mouse decidualization included Wnt4 (Wnt family member 4) [32] (in NPD: fold = 18.2, *p* = 1.45 × 10^−5^, FDR = 0.000162; in IVD: fold = 0.542, *p* = 0.000568, FDR = 0.00359), Egfr (epidermal growth factor receptor) [33] (in NPD: fold = 3.68, *p* = 5.90 × 10^−5^, FDR = 0.000393; in IVD: fold = 0.318, *p* = 1.07 × 10^−5^, FDR = 0.000254), Nr3c1 (glucocorticoid receptor) [34] (in NPD: fold = 2.58, *p* = 5.99 × 10^−5^, FDR = 0.000398; in IVD: fold = 0.456, *p* = 7.26 × 10^−5^, FDR = 0.000877), Foxm1 (forkhead box M1) [35] (in NPD: fold = 8.11, *p* = 3.21 × 10^−6^, FDR = 6.71 × 10^−5^; in IVD: fold = 0.152, *p* = 0.000402, FDR = 0.00284), Spp1 (secreted phosphoprotein 1) [36] (in NPD: fold = 56.6, *p* = 1.79 × 10^−7^, FDR = 1.35 × 10^−5^; in IVD: fold = 0.410, *p* = 0.000321, FDR = 0.00243), and Ptch1 (patched 1) [37] (in NPD: fold = 6.09, *p* = 0.000380, FDR = 0.00147; in IVD: fold = 0.409, *p* = 0.000660, FDR = 0.00401). Nevertheless, we would like to note that pure endometrial stromal cells are used in IVD, while the whole uterus is used for NPD. The whole uterus consists of many cell types, including epithelial cells, stromal cells, endothelial cells, and various immune cells. Differentially expressed genes identified in our NPD model may attribute to non-stromal cells, e.g., immune cells. Thus, our comparative analysis might exaggerate the difference between NPD and IVD.

In humans, the addition of cAMP analogs can shorten the time of the induction of IVD. To boost the IVD process in mice, we treated mouse endometrial stromal cells with 0.5 mM 8-Br-cAMP and 1 μM MPA for 4 days. Mouse decidualization marker genes, Prl8a2, Alpl, Ptgs2 and Bmp2, were examined by qRT-PCR. We found that although Prl8a2 was up-regulated, the other 3 marker genes were down-regulated, suggesting that this regimen is no better than E_2_+P_4_ (Appendix A). Decidualization happens without an embryo in humans; however, this process is embryo-dependent in mice. It seems that the fetus or another stimulus is required and E_2_+P_4_ (or even 8-Br-cAMP+MPA) is inadequate to induce IVD in mice. We conclude that IVD cannot be considered as a reliable model of mouse decidualization. We suggest that the IVD model should be optimized to mimic NPD at the transcriptomic level. Only then can proper insight into the molecular mechanisms underlying decidualization be obtained by using IVD.

## 5. Conclusions

In this study, by using RNA-seq, we compared the global gene expression profiles of 3 mouse models of decidualization. Our study contributes to an increase in the knowledge about mouse models of decidualization.

## Figures and Tables

**Figure 1 genes-11-00935-f001:**
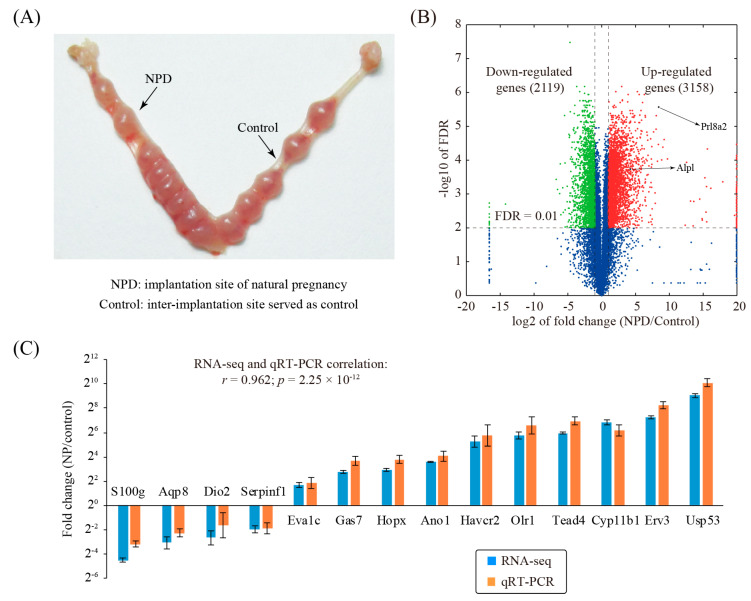
Identification of changed genes in natural pregnancy decidualization (NPD). (**A**) A photograph showing the mouse uterus from day 8 of natural pregnancy. (**B**) Volcano plot showing the comparison between the implantation sites of natural pregnancy decidualization (NPD) and the non-decidualized inter-implantation sites (control) in mice. The cutoff values for differentially expressed genes were: fold change ≥2 and FDR ≤ 0.01. Red, green, and blue colors are indicative of up-regulated genes, down-regulated genes, and non-changed genes, respectively. (**C**) Validation of a panel of selected genes by using qRT-PCR. Data obtained from qRT-PCR were displayed as mean ± SEM (standard error of mean). According to the t-test, *p* < 0.05 is true for all genes. *n* = 3.

**Figure 2 genes-11-00935-f002:**
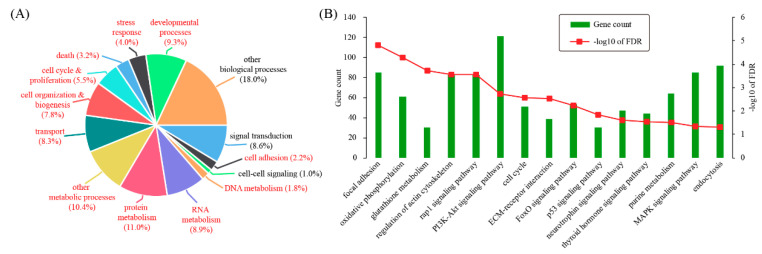
Gene ontology (GO) and pathway enrichment test for differentially expressed genes. (**A**) GO analysis for differentially expressed genes. Differentially expressed genes were grouped according to MGI GOslim terms within biological process categories. GO terms with *p* < 0.05 were colored in red. (**B**) Pathway analysis for differentially expressed genes. The enrichment analysis was carried out by using the DAVID software and the significance threshold for FDR was 0.05.

**Figure 3 genes-11-00935-f003:**
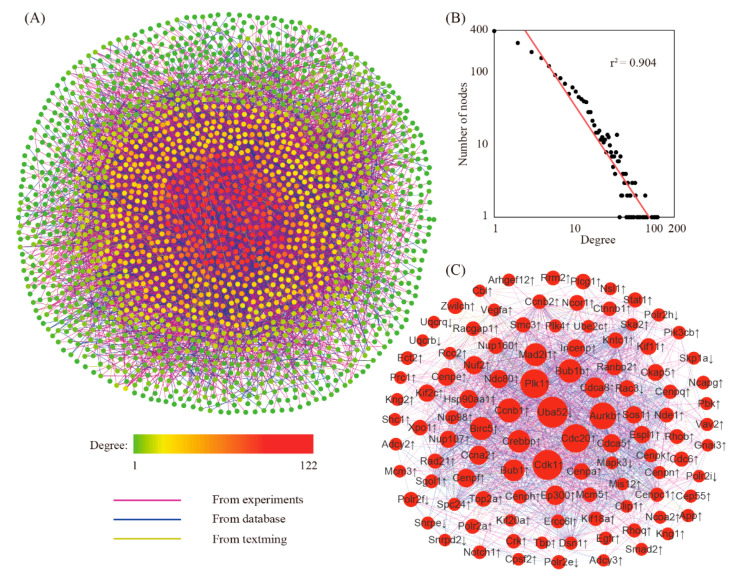
The gene network for differentially expressed genes. (**A**) The structure of the gene network. (**B**) Degree distribution analysis. (**C**) The sub-network focusing on hub genes. Genes with a degree value exceeding mean + 2 × SD were defined as hub genes. ↓, down-regulated genes; ↑, up-regulated genes.

**Figure 4 genes-11-00935-f004:**
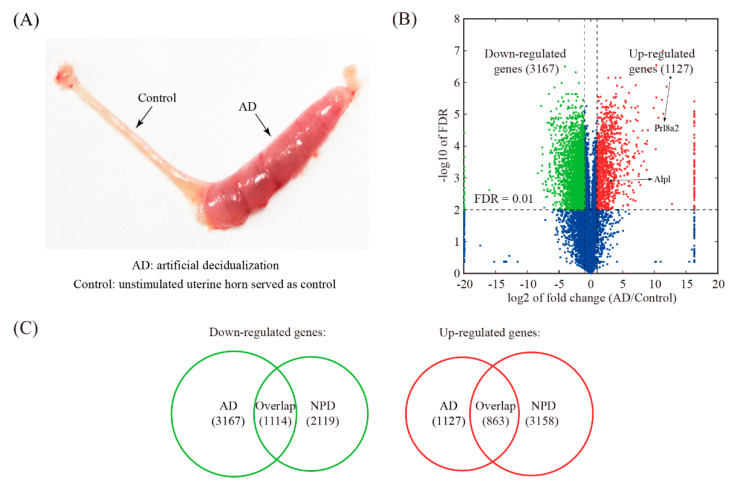
Identification of changed genes in artificial decidualization (AD) model. (**A**) A view of mouse uterus undergoing AD. (**B**) Volcano plot showing the comparison between the oil-stimulated uterine horn (decidualized) and the unstimulated one (served as a control) in mice. The threshold values for differentially expressed genes were: fold change ≥2 and FDR ≤ 0.01. (**C**) Venn diagram showing the overlap of down-regulated genes and up-regulated genes and between the AD model and the NPD model of mouse decidualization.

**Figure 5 genes-11-00935-f005:**
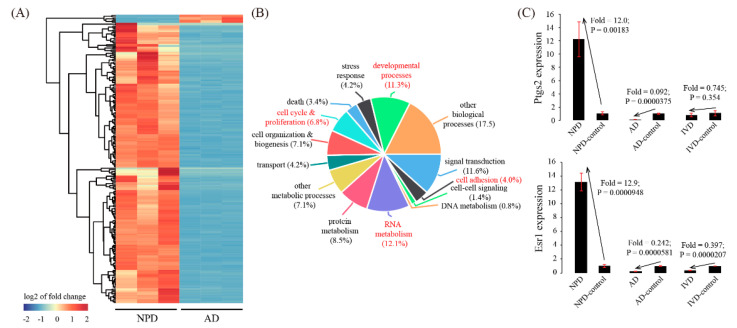
Identification of inconsistently expressed genes in the AD model compared to the NPD model. (**A**) Cluster dendrogram of inconsistently expressed genes. The average linkage clustering algorithm with the Pearson correlation distance measure was used. (**B**) GO enrichment analysis of inconsistently expressed genes. (**C**) Validation of representative inconsistently expressed genes by using qRT-PCR. Data were plotted as mean ± SEM. *n* = 3.

**Figure 6 genes-11-00935-f006:**
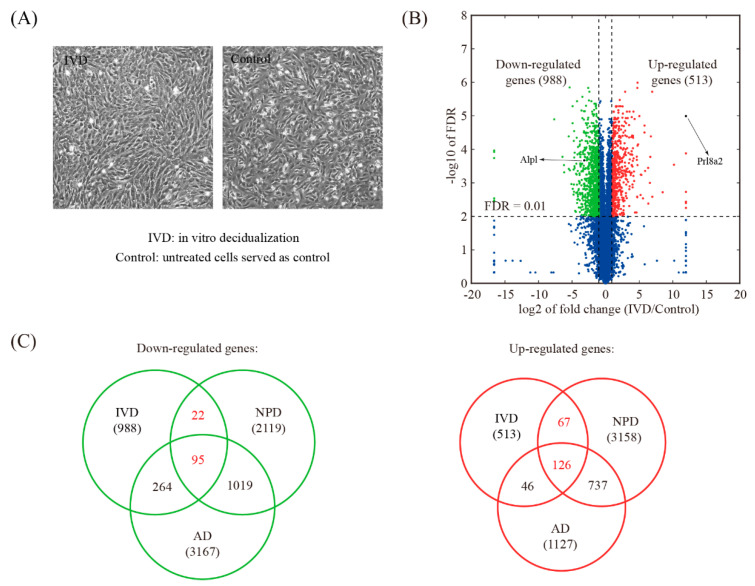
Identification of changed genes in the in vitro decidualization (IVD) model. (**A**) A view of morphological changes in cultured endometrial stromal cells during the IVD process. (**B**) Volcano plot showing the comparison between IVD and control. The threshold values for differentially expressed genes were: fold change >2 and FDR < 0.01. (**C**) Venn diagram showing the overlap of down-regulated genes and up-regulated genes and between the NPD model, the AD model, and the IVD model.

**Figure 7 genes-11-00935-f007:**
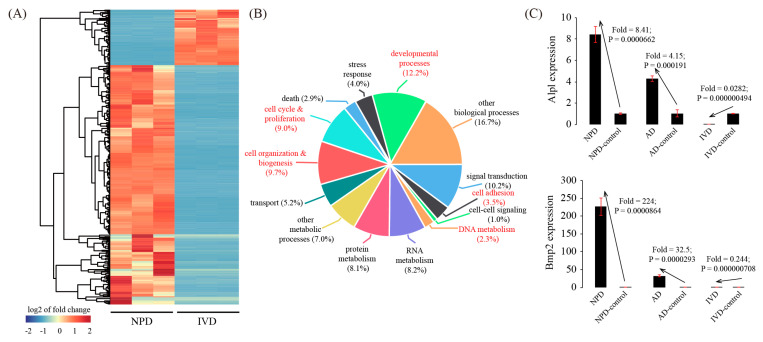
Inconsistently expressed genes in in vitro decidualization (IVD) model compared to NPD. (**A**) Cluster dendrogram of inconsistently expressed genes. The average linkage clustering algorithm with the Pearson correlation distance measure was used. (**B**) GO enrichment analysis of inconsistently expressed genes. (**C**) Validation of representative inconsistently expressed genes by using qRT-PCR. Data were plotted as mean ± SEM. *n* = 3.

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
