# Peer review of "Comparative Analysis of Mouse Decidualization Models at the Molecular Level"

_genes, 2020, doi:10.3390/genes11080935_

Round 1

Reviewer 1 Report

Re: Genes -878678 Comparative analysis of mouse decidualization models at the molecular level

Wang et al. compared common and differentially expressed genes among natural pregnancy decidualization (NPD), artificial decidualization (AD) and in vitro decidualization (IVD) using RNA-seq analysis. They found that expression of 1977 genes and 310 genes are consistent in AD and IVD process, respectively, compared to those expressed in NPD process. Moreover, 217 genes and 456 genes were differentially expressed in in AD and IVD process, respectively, compared to NPD process. Authors conclude that IVD process needs to be optimized for molecular studies of decidualization process. These observations are original. However, this reviewer has some comments to improve the quality of paper.

  • My main concern is that IVD process as well as the time in this paper is not well established. IVD process requires E2 and P4 and cyclic AMP for at least 7 -10 days incubation. Moreover, in vitro decidualization process has to be confirmed by increased prolactin as well as IGFBP1 expression. Thus, one can expect that in the current manuscript use of 4 days of IVD process without cyclic AMP cause a weak decidualization, which can be the main reason why much lower numbers of genes in IVD are affected compared to NPD or AD. Thus, this reviewer first suggests that mRNA and protein levels of prolactin and IGFBP1 needs to be compared among three experimental groups and their controls.
  • Secondly, potential weak in vitro decidualization issue needs to be included in the discussion section.
  • Results from three different decidualization process of RNA-seq analyses need to be presented by triple Venn diagrams to show all common affected up regulated or down regulated genes.

Reviewer 2 Report

            In this manuscript, the authors report a transcriptomic comparison of three methods of mouse decidualization: natural pregnancy (NP), sesame oil-induced deciduoma formation (AD),  and endometrial stromal cells decidualized in vitro with estradiol and progesterone for four days (IVD). The authors evaluate the adequacy of the two less “natural” model systems, and from a comparison of shared upregulated and downregulated genes between the states (respective to an un-decidualized internal control), conclude that the deciduoma model is an acceptably accurate representation of normal decidualization, whereas in vitro decidualization is not.

            Overall, the authors have conducted an interesting RNA-seq comparison of different models of mouse decidualization which is a valuable contribution to reproductive biology and would help inform future experimental practice. However, the manuscript there are a few concerns that need to be addressed before the paper can be published.

The most valuable comparison is between the NP and the deciduoma experiments, since both are RNAseq data from whole uteri, at implantation sites. The comparison to in vitro decidualization (IVD) is more problematic for the following reasons:

  1. As mentioned the natural pregnancy and deciduoma data is based on whole uterus transcriptomes and thus include gene expression from various cell types not represented in in vitro decidualization. For that reason the comparison of natural pregnancy and IVD is invalid. One can not say that IVD is less representative because it does not compare the same kinds of cells.
  2. The IVD protocol used by the authors is E2+P4 treatment for four days. It is highly questionable that this treatment is sufficient to cause decidualization in cultured endometrial stromal cells (ESF). While one can make an argument that E2 and P4 should be sufficient for human ESF decidualization, because in women decidualization happens without an embryo. But even it humans it is questionable whether hormones only treatment really leads to decidualization and if so it usually required >8days of treatment. In mice, however, the fetus or another stimulus is certainly necessary and thus a E2+P4 only is likely inadequate. In humans it is clear that we also need an activation of the PKA pathway, and it is likely that the same is true for mice. There are protocols for mice that also use cAMP, so this would have been a better way to go. In any case it is not justified to say that “in general” the IVD is not a good model of decidual cell differentiation, given the limitations of this study.

My recommendation is to either drop the IVD part of the paper and focus the paper on the comparison between NP and AD, or redo the in vitro experiments with cAMP or some other way to activate the PKA pathway in addition to steroid signaling.

Round 2

Reviewer 1 Report

thank you for providing me the new version. 

This manuscript is a resubmission of an earlier submission. The following is a list of the peer review reports and author responses from that submission.